# On the Decrease in Transformation Stress in a Bicrystal Cu-Al-Mn Shape-Memory Alloy during Cyclic Compressive Deformation

**DOI:** 10.3390/ma14164439

**Published:** 2021-08-08

**Authors:** Tung-Huan Su, Nian-Hu Lu, Chih-Hsuan Chen, Chuin-Shan Chen

**Affiliations:** 1Department of Civil Engineering, Civil Engineering Department Building, National Taiwan University, Room 205, No.1, Sec. 4, Roosevelt Road, Taipei 10617, Taiwan; caedbwind95@caece.net; 2Department of Mechanical Engineering, National Taiwan University, No. 1, Sec. 4, Roosevelt Road, Taipei 10617, Taiwan; f06522712@ntu.edu.tw; 3Department of Materials Science and Engineering, National Taiwan University, No. 1, Sec. 4, Roosevelt Road, Taipei 10617, Taiwan

**Keywords:** full-field stress and strain measurements, shape memory alloys, digital image correlation, data-driven identification, superelasticity, functional fatigue

## Abstract

The evolution of the inhomogeneous distribution of the transformation stress ($\sigma_{s}$) and strain fields with an increasing number of cycles in two differently orientated grains is investigated for the first time using a combined technique of digital image correlation and data-driven identification. The theoretical transformation strains ($\varepsilon_{T}$) of these two grains with crystal orientations ${\left[5\;3\;26\right]}_{\beta }$ and ${\left[6\;5\;11\right]}_{\beta }$ along the loading direction are 10.1% and 7.1%, respectively. The grain with lower $\varepsilon_{T}$ has a higher $\sigma_{s}$ initially and a faster decrease in $\sigma_{s}$ compared with the grain with higher $\varepsilon_{T}$. The results show that the grains with higher $\sigma_{s}$ might trigger more dislocations during the martensite transformation, and thus result in greater residual strain and a larger decrease in $\sigma_{s}$ during subsequent cycles. Grain boundary kinking in bicrystal induces an additional decrease in transformation stress. We conclude that a grain with crystal orientation that has high transformation strain and low transformation stress (with respect to loading direction) will exhibit stable transformation stress, and thus lead to higher functional performance in Cu-based shape memory alloys.

## 1. Introduction

Superelastic shape-memory alloys (SMAs) are functional materials capable of sustaining a large recoverable deformation strain as a result of a stress-induced martensitic transformation (MT) between austenite and martensite. Among SMAs, Cu-Al-Mn SMAs possess superior features such as low cost, high cold workability, and large transformation strain compared with TiNi-based SMAs [1]. They are considered to be suitable candidate materials for a variety of applications ranging from civil engineering to the space industry, in which the SMAs are subjected to cyclic loading. However, the issue of SMA fatigue and fracture is challenging because fatigue problems in Cu-Al-based SMAs are mainly attributed to the constraints of grain boundaries during MT and its resulting plastic deformation [2]. Several studies have extensively investigated the prevention of intergranular fracture problems caused by high elastic anisotropy in the Cu-Al-Mn alloys with the aim of enhancing the superelasticity of polycrystalline Cu-based shape memory alloys [3,4,5]. Therefore, treatments of the microstructure designed to increase grain size, such as the introduction of texture and the reduction of triple junctions, significantly improve the functional performance of Cu-Al-Mn SMAs [6,7,8].

Recently, Cu-Al-Mn single crystals with excellent superelasticity have been fabricated using abnormal grain growth (AGG) induced by a cyclic heat treatment [9,10]. These AGG methods also enable the preparation of large bicrystal samples for mechanical tests, providing insight into the inhomogeneous MT phenomenon and the elastocaloric effect of the sample under compression [11]. The experimental results [11] demonstrate the generation of microcracks at the grain and twin boundaries of the bicrystal Cu-Al-Mn sample under cyclic compression. Although compressive deformation is preferred as a deformation mode to delay the fatigue fracture of metallic materials [12,13], much less is known about the compressive fatigue behavior of Cu-Al-Mn SMAs during cyclic phase transformation.

Based on the above-mentioned motivations and the knowledge of large differences in superelasticity properties of the bicrystal Cu-Al-Mn sample [11], it was expected that such differences in superelasticity properties would cause functional instability of the Cu-based bicrsytal sample during cyclic compressive deformation. As Cu-Al-Mn SMAs are regarded as potential candidates of functional materials, the correlations between strain field and transformation stress, and the evolutions of stress and strain distributions during cyclic superelastic deformation, are critical factors for the compressive fatigue behavior of Cu-Al-Mn SMAs. In this study, we investigated the cyclic compressive behavior of the superelasticity of macro-scale Cu-Al-Mn bicrystals using the digital image correlation (DIC) technique and the data-driven identification (DDI) method. Both methods are used to determine the distributions of transformation stress and strain in the bicrystal and near the grain boundary. Based on the full-field measurement results, the correlations between the decrease in transformation stress, accumulation of residual strain, and martensite transformation are determined.

## 2. Materials and Methods

Figure 1 illustrates the methods employed to characterize the cyclic behavior of the superelasticity of bicrystal Cu-Al-Mn SMAs. The strain and stress fields of a Cu-Al-Mn shape-memory bicrystal were measured using the DIC technique and DDI method, respectively. In this work, we used the same Cu-Al-Mn bicrystal sample prepared in our previous study [11], with dimensions of 8 mm × 4.2 mm × 4.2 mm. Please note that the specimen was subjected to five compression–unloading tests with a maximum global deformation strain from 1% to 5% in 1% increments for each test (see Figure 1 in the previous study [11]). It was found that some plastic deformation occurred when the global deformation strain was higher than 4%. In this work, we further conducted twenty compression–unloading tests using the same Cu-Al-Mn bicrystal sample, which has already undergone the five compression–unloading tests mentioned above. The grain boundary within the sample is indicated by a dashed line in Figure 1. The crystal orientations of both grains were determined via electron backscatter diffraction (EBSD, Oxford Instruments, Abingdon, UK), as shown in the inset of Figure 2a. The thermal analyses of the sample were conducted in a differential scanning calorimeter (DSC, DSC 25, TA Instrument, New Castle, DE, USA) with cooling and heating rates of 10 °C/min. The microstructures were observed by transmission electron microscope (TEM, FEI Tecnai™ G2 F30, Hillsboro, OR, USA) operated at 300 kV. The sample was mechanically ground to a thickness of about 70 μm and then electropolished at −40 °C using HNO_3_ and CH_3_OH (2:8 in volume). The preparation procedures for the bicrystal Cu-Al-Mn sample were detailed in the literature [11].

The cyclic compression–unloading test was performed under the strain-controlled mode using a universal tester with a 50 kN load cell (AG-IS 50 KN, Shimadzu, Japan). The strain rate used in the compression–unloading test was about 2.4 × 10^−3^ s^−1^ such that experiments can be considered as quasi-static. Each compression cycle took about 120 s. A speckle pattern was applied on the observed surface (i.e., area of interest, AOI) of the specimen using black and white sprays. The pattern was used for in situ strain tracing and ex post strain field analysis. The deformation strain of the specimen was measured with a virtual strain gauge by optical DIC (VIC-Gauge 3D, Correlated Solutions, Irmo, SC, USA). Three deformation strains (i.e., global gauge strain $\varepsilon_{g}$ as shown in Figure 1 and the local strain gauges at the top and bottom grains $\varepsilon_{t}$ and $\varepsilon_{b}$ as shown in inset of Figure 2b) were measured using the virtual strain gauge technique. Notably, regardless of the residual strain, a 5% strain (relative to each unloaded state) was applied to the sample during each compression cycle. During the compression test, images of the deformed sample were taken at a rate of 5 Hz using two cameras. Around 600 snapshots were taken for each compression cycle. Because the imaging rate (5 s^−1^) is higher than the strain rate (2.4 × 10^−3^ s^−1^), the deformation behavior of the material can be captured. These snapshots were analyzed ex post in the VIC 3D 8 software to obtain the strain distribution at the surface of the specimen, as shown in the full-field strain measurement in Figure 1.

Based upon the measured results (i.e., DIC data items) of the strain fields and the applied loads from the full-field strain measurement, the equilibrated stress distribution at the surface of the specimen can be numerically computed using the DDI method [14,15,16] under a finite-element framework without the use of constitutive modeling, as illustrated in the full-field stress measurement in Figure 1. For a given set of compression–unloading cycle data, the DDI method uses the governing equations (i.e., stress equilibrium equations) to compute the mechanical stresses at the surface of the specimen. Then, the method is used to identify a database of material states that samples the mechanical stress–strain pairs of material to satisfy the criterion of minimum distance between mechanical stress–strain pairs and material database. Mathematically, this process can be formulated as a constrained minimization problem, which was originally proposed in [14]. The DDI method was validated with synthetic data for linear and non-linear elasticity [14] and was further applied experimentally using real experimental data (i.e., DIC measurements) for elastomer sheet [15] and Cu-Al-Mn shape memory alloy single crystal [16]. In the DDI method, the only governing equations used to determine the stress components are the stress equilibrium equations. Thus, the bias introduced by the choice and the calibration of a constitutive model was removed. Further details about the full-field stress and strain measurements can be found in the literature [15,16].

The mechanically admissible stress–strain pairs, obtained from the full-field stress and strain measurements shown in Figure 1, are stored at each element and are considered to represent the mechanical state $\left(\sigma_{e},\varepsilon_{e}\right)$, as shown in the stress–strain responses in Figure 1. Using a linear regression analysis of these mechanical states in both the elastic and plateau regions, the distribution of MT stress $\sigma_{s}$ can be obtained from the intersection of the two linear stress–strain curves, as illustrated by the stress–strain responses shown in Figure 1. The residual strain $\varepsilon_{r}$ and transformation strain $\varepsilon_{\mathrm{tr}}$ can be obtained readily, as shown in the stress–strain responses of Figure 1.

## 3. Results

Figure 2a shows the crystal orientations of the top and bottom grains along the loading direction (LD), as determined using EBSD, which were along ${\left[5326\right]}_{\beta }$ and ${\left[6511\right]}_{\beta }$, respectively. The theoretical transformation strain ($\varepsilon_{T}$) of the transition from the β phase to 6M martensite during compression was calculated based on the Wechsler–Liebermann–Read theory [8,11,17]. The values of $\varepsilon_{T}$ were 10.1% and 7.1% for the top and bottom grains, respectively. Note that the compressive transformation strain of these grains significantly differed in the loading direction.

Figure 2b shows the compressive stress–strain curves of both grains when a gauge strain ($\varepsilon_{g}$) of 5% was applied, which covered the entire specimen during deformation, as illustrated in Figure 1. Local virtual strain gauges $\varepsilon_{t}$ and $\varepsilon_{b}$ were used to measure the average strains in the top and bottom grains, respectively (inset of Figure 2b). It can be seen that the stress-induced martensitic transformation (SIMT) of the top grain (blue lines) occurred at transformation stresses of 306 MPa and 297 MPa for the first and twentieth cycles, respectively. By contrast, for the bottom grain (red lines), the transformation stresses of the first and twentieth compression cycles were 398 MPa and 292 MPa, respectively. Both grains exhibited different cyclic behaviors on average, including the decrease in transformation stress, accumulation of irrecoverable strain, and the transformation strain (see Table 1). As shown in Table 1, the properties (i.e., $\sigma_{s}$, $\varepsilon_{r}$, and $\varepsilon_{\mathrm{tr}}$) of the total curve are between those of the top and bottom grains, which were reported in [11]. Please note that the total stress–strain curve is denoted as “Average” in first column of Table 1. According to the results of residual strain $\varepsilon_{r}$ (sixth column of Table 1), it was found that the plastic deformation of the entire specimen was mainly contributed from the bottom grain. In short, combinations of crystal orientation in bicrystal SMAs will result in varied mechanical properties of the entire specimen.

Figure 3a shows the evolution of the axial strain field $\varepsilon_{\mathrm{yy}}$ while loading toward and unloading away from a gauge strain $\varepsilon_{g}$ of 5% during the first, tenth, and twentieth compression–unloading cycles. The transformation stress fields are shown in Figure 3b, which illustrates the distribution of transformation stress in the specimen. As shown in the first compression cycle, the top grain underwent most of the deformation during the loading process. By contrast, the bottom grain began its partial MT after an $\varepsilon_{g}$ of 3%. The difference in transformation behavior between the top grain and the bottom grain can be ascribed to differences in the MT stresses required to trigger MT, as shown in Figure 3b. The transformation stress of the top grain was approximately 325 MPa, which was less than that of the bottom grain (approximately 400 MPa), indicating that the top grain was more likely to begin MT earlier until the loading force was high enough to initiate MT in the bottom grain. The transformation stresses of both grains near the grain boundary (indicated by white dashed lines) were smaller than those further away from the grain boundary, as shown in Figure 3b, indicating that the stress state around the grain boundary promoted MT at a lower stress level.

Before the beginning of the tenth compression cycle, some regions in the bottom grain had residual strain. At the tenth deformation, the top grain experienced less deformation relative to the first cycle, whereas the bottom grain began to exhibit increased deformation. As shown in Figure 3b, at the tenth cycle, the transformation stress of the upper part of the bottom grain (near the grain boundary) decreased, bringing the values closer to those of the top grain. This decrease in transformation stress resulted in an increase in regions in which the MT could be triggered in the bottom grain, leading to increased transformation strain in the bottom grain.

At the beginning of the twentieth compression cycle, the band of residual strain at the bottom grain extended, and more residual strain remained. Furthermore, during the loading process, the upper part of the bottom grain showed a level of transformation stress closer to that of the top grain, as shown in Figure 3b. According to Figure 3b, the decrease in transformation stress in the bottom grain was initiated around the grain boundary and then propagated to the lower part of the bottom grain.

By comparing the evolution of the strain distributions during the cyclic deformation, it can be deduced that the region exhibiting a decrease in transformation stress was highly correlated with the region undergoing MT. In the first compression test, the MT band in the bottom grain was clearly identified. During cyclic deformation, the MT in the bottom grain mainly originated from this band, and the residual strain in this region accumulated. This band, which was associated with accumulated permanent deformation, also experienced a more severe decrease in transformation stress than that of the top grain, as shown in Figure 3c, which reveals the decrement in transformation stress after twenty cycles. Because the transformation stress in the top and bottom grains (near the grain boundary) became similar after cyclic deformation, a concurrent MT occurred in the later cycles in these grains, leading to a significantly different deformation behavior from that of the first cycle.

To further investigate the relationships between transformation stress ($\sigma_{s}$) and residual strain ($\varepsilon_{r}$), the local axial stress–stain responses ($\sigma_{\mathrm{yy}},\;\varepsilon_{\mathrm{yy}})$ at probing points A, B, and C (Figure 3b) are shown in Figure 4a. The evolutions of the transformation stresses and the accumulations of residual strains at these points are shown in Figure 4b,c. Point A was set in the top grain, and points B and C in the bottom grain were placed in the regions that underwent full MT and partial MT, respectively. At point A in the top grain, a stable transformation stress was observed after twenty compression cycles. By contrast, varied mechanical responses in terms of transformation stress and accumulation of irrecoverable strain were observed for points B and C in the bottom grain. The transformation stress and residual strain at these three points were quantified and are presented in Figure 4b,c, respectively. Figure 4b shows a comparison of different grain orientations (points A and B). The transformation stress at point A was observed to have slight decreasing behavior, while at point B, the decrease in transformation stress was more significant (i.e., from 400 MPa to 311 MPa). The results also show that, for points having the same grain orientation (points B and C), point B, which undergoes more MT (i.e., higher transformation strain, $\varepsilon_{\mathrm{tr}}$), exhibits faster decreasing behavior than point C, as determined from the stress–strain curve in Figure 4a. In other words, in a single grain, a region that underwent more MT (i.e., high $\varepsilon_{\mathrm{tr}}$) experienced a greater decrease in transformation stress. Notably, after twenty cycles, the transformation stresses at points A and B became nearly equal, as shown in Figure 4b, which resulted in more MT in the bottom grain.

Considering the loading cycles, a region that underwent more MT in the bottom grain also caused a faster accumulation of unrecoverable strain, as shown in Figure 4c. An $\varepsilon_{r}$ of 4% at point B was observed after twenty compression cycles, roughly four times greater when compared with the accumulation at point C. For point A, an $\varepsilon_{r}$ of 0.5% was observed, which is the minimum strain among these points owing to its lower transformation stress. In different grains, the grain requiring a higher stress to induce MT (the bottom grain) showed a larger residual strain and faster decrease in transformation stress. In addition, in the same grain (the bottom grain), the regions with more MT accumulated more residual strain and exhibited a clear decrease in transformation stress.

## 4. Discussion

The difference in transformation stress between the two crystal grains is attributed to the difference of their crystal orientations. According to [18], the habit planes of Cu-Al-Mn martensite are {0.16 −0.72 −0.68} and the shear directions are <0.14 −0.65 0.74>. With these transformation systems, the maximum Schmid factor of the two grains with orientations ${\left[5\;3\;26\right]}_{\beta }$ (top grain) and ${\left[6\;5\;11\right]}_{\beta }$ (bottom grain) is determined to be 0.49 and 0.32, respectively. As the Schmid factor of the bottom grain is smaller than that of the top grain, larger stress is needed to trigger MT in the bottom grain. Therefore, the transformation stress of the bottom grain (398 MPa) is larger than that of the top grain (306 MPa), as shown in Figure 2b and Table 1.

The previous results (Figure 3) show that the decrease in transformation stress in a bicrystal Cu-Al-Mn SMA under a strain-controlled cyclic compression–unloading test was affected by the accumulation of residual strain. These macroscopic residual strains are mainly a result of dislocation slips in the austenite phase [19,20,21,22,23,24] and accumulated residual martensite phase owing to an incomplete reverse MT [19,25,26]. These dislocation slips, which can be triggered separately during forward and reverse MTs [27], are fostered by localized stress fields between austenite–martensite interfaces during forward and reverse MTs [20,28,29,30]. Such dislocation slips and residual martensite can also cause mesoscopic residual stress fields within the specimen [31]. Therefore, the mesoscopic residual stress field, which is of the same type as the applied stress, assists in the nucleation of martensite variants [19], and thus leads to a significant reduction in the macroscopic transformation stress required to trigger MT during subsequent cycles [32,33,34,35,36].

In order to provide clear evidence, the bottom grain was cut from the bicrystal sample for thermal analysis. As shown in Figure 5a, after 20 cyclic deformations, the first heat curve shows that the reverse MT occurred at about 140.1 °C. The sample was then cooled to −140 °C (Step 2), and a forward MT was identified at −87.7 °C. During the second heating (Step 3), the reverse transformation occurred at −69.7 °C, instead of the 140.1 °C in the first heating curve. This feature indicated that the martensite was stabilized during the cyclic compressions. The stabilized martensite needed a higher temperature to transform back to austenite, as shown in the first heating curve. In the successive cooling and heating, the martensite was thermally induced and thus was not stabilized, causing the reverse transformation temperature to be restored to its normal value (−69.7 °C). The feature of martensite being stabilized after deformation was also reported in other Cu-based [37] and TiNi-based SMAs [38].

TEM analyses were performed on the bottom grain after cyclic compression. Figure 5b shows the TEM bright-field image of the bottom grain, in which entangled dislocations can be observed. Figure 5c shows large amounts of residual martensite in the bottom grain, indicating that martensite was stabilized at room temperature by dislocations after cyclic deformation. The TEM observations confirmed that the residual strain is caused by the dislocation and residual martensite formed during cyclic deformation.

As shown in Figure 2b, the higher transformation stress in the bottom grain generates more dislocations in the austenite phase and more residual martensite during the forward and reverse MT. Both mechanisms lead to plasticity formed mainly in the bottom grain. Both the dislocation and stabilized martensite can cause mesoscopic residual stress fields within the specimen. Such residual stress assisted the nucleation of martensite variants and thus led to a significant reduction in the macroscopic transformation stress required to trigger MT during subsequent cycles. On the other hand, the top grain exhibited smaller residual strain owing to its smaller transformation stress, and thus its transformation stress remained stable. In addition, in the bottom grain, the decreasing behavior varied significantly between regions undergoing different levels of transformation strain (points B and C). As shown in Figure 3c or Figure 4b, point B experienced more MT and associated residual strain compared with point C. Consequently, point B exhibited a larger decrease in transformation stress than point C. These results support the hypothesis that the mesoscopic stress field caused by dislocations or residual martensite assisted MT.

In this study, the horizontal strain fields $\varepsilon_{\mathrm{xx}}$ at the surface of the specimen were examined to investigate the deformation in the vicinity of the grain boundary. Figure 6a shows the evolution of $\varepsilon_{\mathrm{xx}}$ during loading toward and unloading away from a gauge strain $\varepsilon_{g}$ of 5% for the first, tenth, and twentieth compression–unloading cycles. As shown in the unloaded state ($\varepsilon_{g}$ = 0.2%) of the first compression cycle, the bottom grain exhibits an accumulation of residual $\varepsilon_{\mathrm{xx}}$ near the grain boundary (highlighted in the red rectangle), while the remaining part of the grain boundary shows relatively less residual $\varepsilon_{\mathrm{xx}}$. In the tenth and twentieth cycles, this accumulation of residual $\varepsilon_{\mathrm{xx}}$ in the red rectangle kept increasing and expanding toward the remaining part of the grain boundary. Note that, in the right half-part of the grain boundary, the bottom grain accumulated more residual $\varepsilon_{\mathrm{xx}}$ than the top grain. These differences in the accumulation of residual $\varepsilon_{\mathrm{xx}}$ or strain incompatibility in the vicinity of the grain boundary in a bicrystal reveal the relative movement between the top grain and bottom grain.

To further investigate incompatibility conditions of the grain boundary, we quantified the average incompatibility strain $∆\varepsilon_{\mathrm{xx}}^{\mathrm{avg}}$ near the grain boundary. The definition of $∆\varepsilon_{\mathrm{xx}}^{\mathrm{avg}}$ is the average $\varepsilon_{\mathrm{xx}}$ in the selected region of the bottom grain minus the average $\varepsilon_{\mathrm{xx}}$ in the selected region of the top grain. As shown in inset of Figure 6b, we selected four regions near the grain boundary (R1, R2, R3, and R4) and paired the regions (i.e., R2–R1 and R4–R3) to compute their evolution of $∆\varepsilon_{\mathrm{xx}}^{\mathrm{avg}}$ regarding the selected compression–unloading cycles, C1, C10, and C20. In Figure 6b, the average incompatibility strain $∆\varepsilon_{\mathrm{xx}}^{\mathrm{avg}}$ in the region R4–R3 increased from 0.07% to 1.51%, while the $∆\varepsilon_{\mathrm{xx}}^{\mathrm{avg}}$ in the region R2–R1 increased from 0.05% to 0.67%. These results show that the incompatibility of the grain boundary in the region R4–R3 is more severe than that in the region R2–R1. Such a difference between these two regions can be ascribed to the angle between the loading direction and normal direction of the grain boundary. As can be seen in the inset of Figure 6b, the grain boundary in the region R4–R3 deviates from the horizontal plane by approximately 22.5 degrees. This variation in the direction of grain boundary will introduce the major axial deformation $\varepsilon_{\mathrm{yy}}$ as an extra component of tangential movement to the inclined grain boundary (see the right half-part of the grain boundary during the loading process in Figure 3a). Hence, the regions near the inclined grain boundary will experience not only larger axial deformation (Figure 3a), but also larger relative movement (Figure 6a) compared with the regions near the flat grain boundary. Consequently, a significant difference in average incompatibility strain $∆\varepsilon_{\mathrm{xx}}^{\mathrm{avg}}$ between the two regions (i.e., R2-R1 and R4-R3) occurs, as shown in Figure 6b. Additionally, as reported in the previous work [11], the top and bottom grains underwent outward and inward out-of-plane deformations during compression–unloading cycles. This out-of-plane motion was considered for causing generation of microcracks. In this study, we found that the incompatibility along the x direction of the bicrystal Cu-Al-Mn sample under cyclic compression could be another mechanism for formation of microcrack at the grain boundary. Thus, even though compressive deformation is considered a preferred deformation mode to delay fatigue fracture of metallic materials [12,13], the out-of-plane motion and relative deformation near the grain boundary along the x direction in a bicrystal Cu-Al-Mn sample may cause cracking and even fracture in Cu-Al-Mn SMAs.

With the aid of the full-field stress and strain measurements (i.e., DIC and DDI techniques) for revealing strain and transformation stress fields, it was found that grain boundary kinking plays an important role in the decrease in transformation stress near the grain boundary in the bicrystal Cu-Al-Mn sample. As can be seen in Figure 3a, during the loading process, because the right half-part of the grain boundary underwent more MT than the other part of the grain boundary, more residual strain remained near the right half-part of the grain boundary at the end of the cycle ($\varepsilon_{g}$ = 0.2% in Figure 3a). As mentioned above, residual strain causes the decrease in transformation stress. Hence, this additional residual strain near the kink grain boundary will induce an extra decrease in transformation stress. As can be seen in Figure 3c, the transformation stress near the right half-part of the grain boundary degraded faster than other part of the grain boundary. Thus, we concluded that the grain boundary kinking in bicrystal induces an additional decrease in transformation stress.

Furthermore, during the strain-controlled cyclic loading (5% strain for each cycle), the transformation stress of the bottom grain gradually decreased to the values close to those of the top grain (Figure 3b,c). Hence, the bottom grain gradually had more MT, and thus a larger $\varepsilon_{\mathrm{tr}}$ than in its first cycle, as shown in Figure 3a. This explains why the average $\varepsilon_{\mathrm{tr}}$ in the bottom grain increased with the increasing numbers of deformation cycles, as shown in Figure 2b and Table 1. By contrast, because the bottom grain contributed more deformation, the MT and the associated strain contributed by the top grain decreased when the deformation cycles increased, as shown in Figure 2b or Figure 3a. Hence, the average $\varepsilon_{\mathrm{tr}}$ in the top grain decreased with the increasing number of deformation cycles, as shown in Table 1. Therefore, the initially inhomogeneous deformation behavior in the bicrystal sample became slightly more homogeneous after twenty cyclic compressive deformations, as shown in Figure 3a.

The full-field stress and strain measurements provide a promising technique for measuring stress–strain responses in SMAs. This new method offers not only insights into cyclic superelastic deformation, but also the compressive fatigue behavior of Cu-Al-Mn SMAs. With this method, several future research directions could be considered, including the influence of misorientation between two grains on its mechanical properties, the effect of grain boundary geometry on the grain boundary strength [39,40,41], and the shape memory recoverability between two grains in a bicrystal sample. Furthermore, a direct connection between macroscopic shape memory effects and stress and strain states at the materials’ grains could also be elucidated in the future.

## 5. Conclusions

In summary, this study investigated the distribution of transformation stress and strain fields in a bicrystal Cu-Al-Mn sample under cyclic compression. The decrease in transformation stress in both grains correlated with the accumulation of residual strain. The accumulation of residual strain depends on factors such as grain orientation along the loading directions, transformation strain, and grain boundaries. These experimental results and analyses demonstrated that dislocation slip and residual martensite were triggered more easily when a higher transformation stress was required to trigger MT, thus resulting in greater residual strain and a larger decrease in transformation stress. The decreasing behavior at the grain boundary was related to strain incompatibility and the angle between loading direction and the normal direction of grain boundary. Consequently, microstructures with low-angle grain boundaries or single crystals, which exhibit more homogenous deformation behaviors and less restrictions from grain boundaries, will demonstrate higher functional stability and thus longer lifetimes during their cyclic service lives.

## Figures and Tables

**Figure 1 materials-14-04439-f001:**
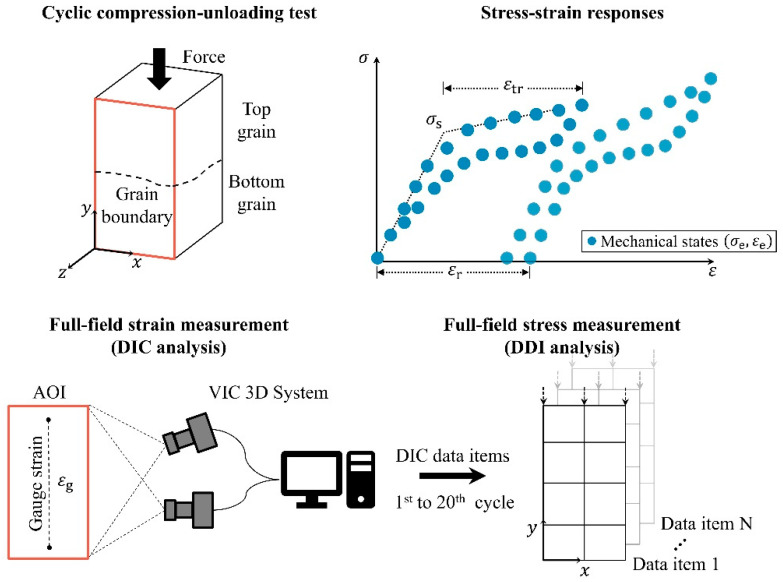
The digital image correlation (DIC) technique and data-driven identification (DDI) method were employed to measure the strain and stress distributions, respectively, at the surface of the specimen to characterize the cyclic behavior of the superelasticity of the bicrystal Cu-Al-Mn SMAs. The cyclic compression–unloading test was performed under the strain-controlled mode. The strain fields in the area of interest (AOI) can be obtained using the DIC technique. Based on the experimentally determined strain fields, the stress fields in the AOI can be computed using the DDI method. Finally, three parameters (i.e., transformation stress ($\sigma_{s}$), residual strain ($\varepsilon_{r}$), and transformation strain ($\varepsilon_{\mathrm{tr}}$)) can be computed from the stress–strain responses.

**Figure 2 materials-14-04439-f002:**
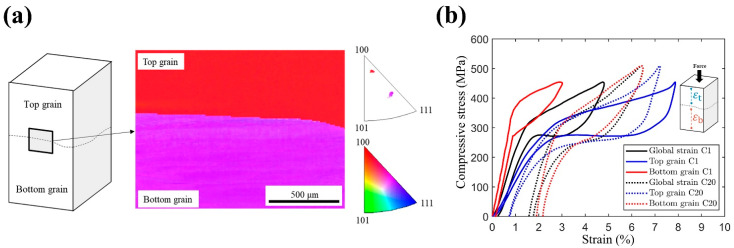
(**a**) Geometry of the bicrystal Cu-Al-Mn SMA. The loading directions of the top and bottom grains are shown in the inverse pole figure. (**b**) Average stress–strain curves of the top grain ($\varepsilon_{t}$), bottom grain ($\varepsilon_{b}$), and the entire specimen ($\varepsilon_{g}$). The bicrystal sample was loaded to a gauge strain ($\varepsilon_{g}$) of 5% during cyclic deformation. Local virtual strain gauges $\varepsilon_{t}$ and $\varepsilon_{b}$ were used to measure the average strains in the top and bottom grains, respectively (inset of (**b**)).

**Figure 3 materials-14-04439-f003:**
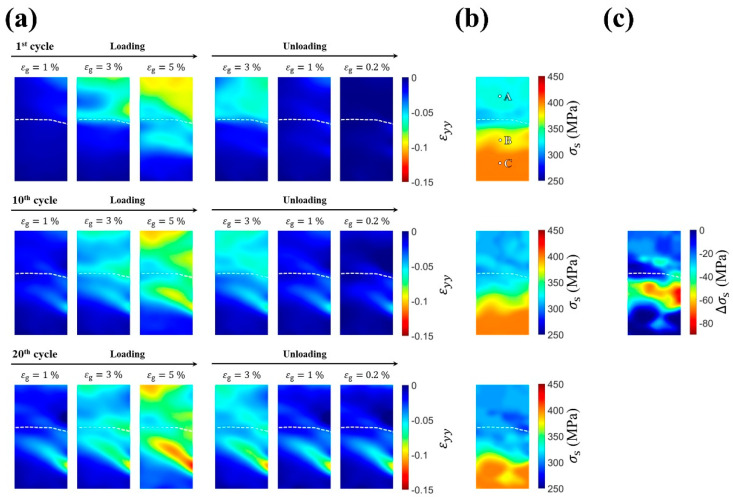
Distribution of (**a**) axial strain fields $\varepsilon_{\mathrm{yy}}$ during loading toward and unloading away from the gauge strain $\varepsilon_{g}$ of 5% and (**b**) transformation stress fields $\sigma_{s}$ in the bicrystal Cu-Al-Mn SMA sample for selected compression–unloading cycles: C1, C10, and C20. Points A, B, and C are probing points for recording the local axial stress–strain responses $\left(\sigma_{\mathrm{yy}},\varepsilon_{\mathrm{yy}}\right)$ as shown in Figure 4a. (**c**) Transformation stress difference $∆\sigma_{s}$, which is the difference in transformation stress between cycles 1 and 20, shown in the plot in Figure 3b.

**Figure 4 materials-14-04439-f004:**
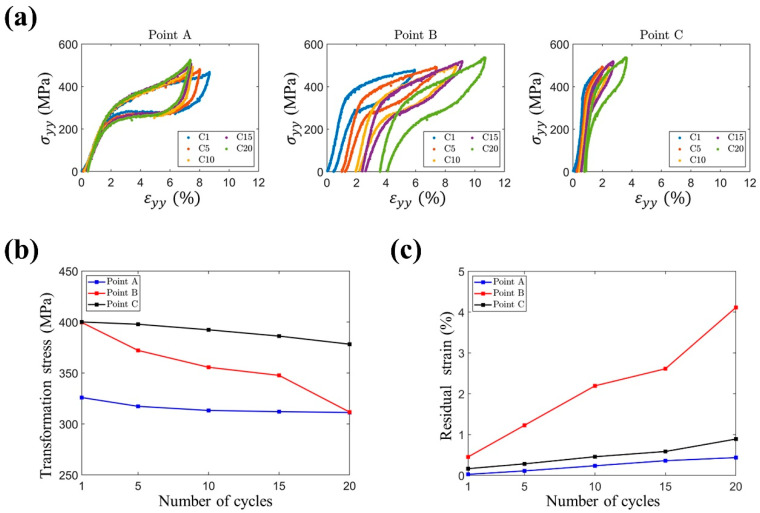
(**a**) Local axial stress–strain responses ($\sigma_{\mathrm{yy}},\;\varepsilon_{\mathrm{yy}})$ recorded by the probing points (according to Figure 3b) along the axial centerline for several selected compression–unloading cycles (C1, C5, C10, and C20). The evolution of the (**b**) transformation stress $\sigma_{s}$ and (**c**) residual strain $\varepsilon_{r}$ with respect to the number of cycles. These values are computed from the local axial stress–strain responses shown in (**a**).

**Figure 5 materials-14-04439-f005:**
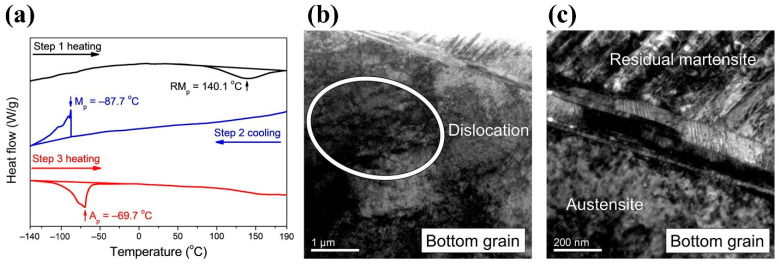
(**a**) Thermal analysis of the bottom grain after 20 compression cycles. (**b**,**c**) TEM bright field images of the bottom grain, which show the formation of dislocations and residual martensite after cyclic compression, respectively.

**Figure 6 materials-14-04439-f006:**
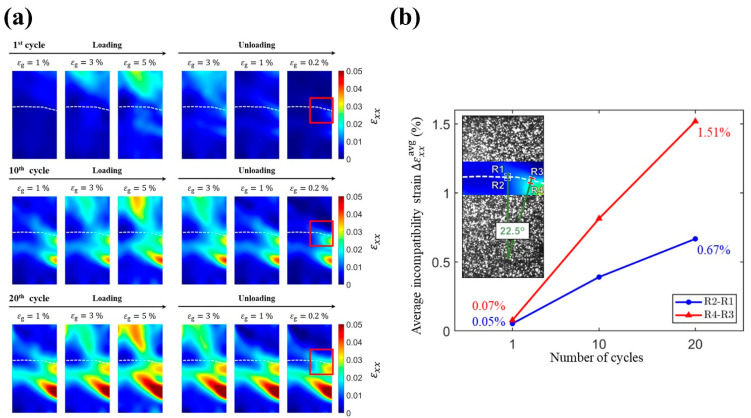
(**a**) Distribution of horizontal strain fields $\varepsilon_{\mathrm{xx}}$ during loading toward and unloading away from the gauge strain $\varepsilon_{g}$ of 5% in the bicrystal Cu-Al-Mn SMA sample for selected compression–unloading cycles: C1, C10, and C20. (**b**) The evolution of average incompatibility strain $∆\varepsilon_{\mathrm{xx}}^{\mathrm{avg}}$ in the regions (R2-R1 and R4-R3) with respect to the number of cycles. These values are computed from the strain fields multiplied by transformation matrix based on the angle between loading direction and normal direction of the grain boundary (inset of (**b**)).

**Table 1 materials-14-04439-t001:** Loading direction (LD), theoretical transformation strain ($\varepsilon_{T}$), transformation stress ($\sigma_{s}$), residual strain ($\varepsilon_{r}$), and transformation strain ($\varepsilon_{\mathrm{tr}}$) of the top grain, bottom grain, and the entire specimen (average) for the first and twentieth compression cycles. Three parameters (i.e., $\sigma_{s}$, and $\varepsilon_{\mathrm{tr}}$) were computed from the stress–strain curves of the top grain, bottom grain, and the entire specimen (average) shown in Figure 2b.

	Loading Direction	Theoretical Transformation Strain (%)	Number of Cycles	Transformation Stress (MPa)	Residual Strain (%)	Transformation Strain (%)
Top	[5 3 26]	10.1	1st	306	0.12	5.7
			20th	297	0.77	5.1
Bottom	[6 5 11]	7.1	1st	398	0.14	2.0
			20th	292	2.17	3.9
Average	−	−	1st	313	0.24	3.5
			20th	290	1.78	4.0

## Data Availability

The data that support the findings of this study are available from the corresponding author upon reasonable request.

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
