# Peer review of "On the Decrease in Transformation Stress in a Bicrystal Cu-Al-Mn Shape-Memory Alloy during Cyclic Compressive Deformation"

_materials, 2021, doi:10.3390/ma14164439_

Round 1

Reviewer 1 Report

  1. It seems to me that it is worth explaining in the text of the manuscript why it is necessary to prepare bicrystal sample with regions of different crystallite orientations. Why were these grain orientations chosen exactly?
  2. For grains in the top of the sample, with an increase in the number of cycles, the transformation strain (Table 1) decreases. Why?

3. According to Fig. 4 the difference between the top and bottom of the sample according to the studied indicators is large enough. Did you achieve this? Or is it a negative result? How does this difference affect the shape memory effect, will there be cracks in the transition layer during martensitic transformation?

Round 2

Reviewer 1 Report

It may be worth adding to the discussion or conclusions section your further directions of research, namely, the influence of grain orientation not only on the arising stresses, but also directly on the shape memory effect.

Reviewer 2 Report

The revisions made to the text are excellent, and clarify all the unresolved issues present in the original version.

I have one concern though, and suggest removing the added discussion on the Clausius-Clapeyron related explanation for the difference in transformation stress between the two grains (lines 267-274). I do not think this is a viable physical explanation. The Clausius-Clapeyron equation provides information on the slope of the transformation stress vs temperature curve, but cannot be used to infer on the different transformation stress values of the two grains at a given temperature. The first explanation regarding the different Schmidt factors is valid and sufficient. 
